# Towards Flawless Designs: Recent Progresses in Non-Orthogonal Multiple Access Technology

**Gaoyuan Dai** [1,2,†], **Ronglan Huang** [1,†], **Jing Yuan** [3], **Zeng Hu** [4], **Longru Chen** [2], **Jianxian Lu** [2], **Tianrui Fan** [2], **Dehuan Wan** [5,*], **Miaowen Wen** [1,*], **Tianwei Hou** [6,7] and **Fei Ji** [1]

1 School of Electronic and Information Engineering, South China University of Technology, Guangzhou 510641, China
2 Guangdong Branch of China Telecom Co., Ltd., Guangzhou 510180, China
3 School of Information Engineering, East China Jiaotong University, Nanchang 330013, China; jyuan@ecjtu.edu.cn
4 College of Information Science and Technology, Zhongkai University of Agriculture and Engineering, Guangzhou 510225, China
5 Center for Data Science and Artificial Intelligence, Guangdong University of Finance, Guangzhou 510521, China
6 School of Electronic and Information Engineering, Beijing Jiaotong University, Beijing 100044, China
7 Institute for Digital Communications, Friedrich–Alexander Universität Erlangen–Nürnberg (FAU), 91054 Erlangen, Germany
* Correspondence: wan_e@gduf.edu.cn (D.W.); eemwwen@scut.edu.cn (M.W.)
† These authors contributed equally to this work.

**Abstract:** High effectiveness and high reliability are two fundamental concerns in data transmission. Non-orthogonal multiple-access (NOMA) technology presents a promising solution for high-speed data transmission, which has long been pursued by academia and industry. However, there is still a significant road ahead for it to effectively support a wide range of applications. This paper provides a comprehensive study, comparison, and classification of the current advanced NOMA schemes from the perspectives of single-carrier (SC) systems, multicarrier (MC) systems, reconfigurable-intelligent-surface (RIS)-assisted systems, and deep-learning (DL)-assisted systems. Specifically, system implementation issues such as the transition from SC-NOMA to MC-NOMA, the relaxation of distinct channel gains, the consideration of imperfect channel knowledge, and the mitigation of error propagation/intra-group interference are involved. To begin with, we present an overview of the state-of-the-art developments related to the advanced design of SC-NOMA. Subsequently, a generalized MC-NOMA framework that provides the diversity–multiplexing gain by enhancing users' signal-to-interference-plus-noise ratio (SINR) is proposed for better system performance. Moreover, we delve into discussions on RIS-assisted NOMA systems, where the receiver's SINR can be enhanced by intelligently reconfiguring the reflected signal propagations. Finally, we analyze designs that combine NOMA/RIS-NOMA with DL to achieve highly efficient data transmission. We also identify key trends and future directions in deep-learning-based NOMA frameworks, providing valuable insights for researchers in this field.

**Keywords:** non-orthogonal multiple-access (NOMA); single-carrier paradigm; multicarrier framework; multiplexing and diversity; reconfigurable intelligent surface (RIS); deep learning (DL)

## 1. Introduction

The evolution of modern communications technologies, with the developments of the last ten years, has changed from connecting people to providing connections for people, things, the Earth, and even the universe. Therefore, enabling each device to enjoy ubiquitous connectivity is the inevitable trend of future communications and networking. The Fifth-Generation (5G) network, for example, opens the era that everything could be connected, and this story line will be enriched and extended by its successors such as the

Sixth-Generation (6G) or the Seventh-Generation (7G) network to realize the great vision of "One as All, All as One" [1–3].

In fact, the networking construction of connecting everything is already in progress. According to the report from Market Research Future, the machine-to-machine connections market size was valued at USD 35.6 billion in 2022. The machine-to-machine connections market industry is projected to grow from USD 37.5 billion in 2023 to USD 57.4 billion in 2032, exhibiting a compound annual growth rate of 5.40% during the forecast period (2023–2032). On the other hand, and coincidentally, Cisco's predictions were similar to the above prediction, in which it reported that about 14.7 billion machine-to-machine devices would be deployed to support the Internet of Things (IoT) by the year 2023, and more than 78 billion devices are connected via cellular networks to access services. Furthermore, with the explosive growth of devices/equipment intended for the IoT, it has been reported that an average of 6–7 devices were carried per person by the year 2020, and the number of devices connected per person will be about 20.5 billion by the year 2030. Besides, there are millions of mobile users with an annual growth rate of around 25%, and the number is expected to reach 80 billion by 2030 [4,5].

How to effectively support such a huge amount of IoT networking devices and meet their explosive data traffic requirements are of great importance, yet represent challenging tasks [6]. Fortunately, this issue can be effectively addressed with the utilization of non-orthogonal multiple-access (NOMA) techniques. NOMA surpasses the limitations of orthogonal multiple-access (OMA) techniques and offers appealing advantages that are highly desirable in future networks, including vehicular communication networks. These advantages include higher-speed data exchange, improved reliability, denser connectivity, and reduced latency [7,8]. Therefore, one of the motivations of this paper was to summarize methods for suppressing intra-group interference in NOMA. Additionally, studies on how to effectively relax the stringent requirements on users' channel qualities and reduce the computational complexity for users' symbol decoding to achieve performance gains in NOMA were thoroughly reviewed. Specifically, NOMA provides a compelling solution to meet the evolving demands of modern networks [9,10]. As a member of the family of multiple-access techniques, the idea of NOMA dates far back in the past, with the first being introduced to wireless communications systems coming as early as 1972. After 2014, it suddenly gained momentum and became a hot research topic in the field of multiple-access techniques, and one can see that the cumulative results of yearly searches for publications using the acronym "NOMA" on the IEEE Xplore Digital Library over the past decade include a sharp increase in conference papers and journal publications, as illustrated in Figure 1. Besides, it can integrate with other existing key techniques, for example massive multiple-input multiple-output (massive MIMO) [11–17], cooperative communication [18–21], millimeter wave (mmWave) [22,23], index modulation (IM) [10,24–26], the reconfigurable intelligent surface (RIS) (also termed the intelligent reflecting surface (IRS)) [27–34], artificial intelligence (AI) [35–42], edge computing [43,44], and holographic technology [45], to strive for further improvement of system performance. With the ubiquity and pervasiveness of IoT devices in our daily activities, in a word, NOMA is currently attracting more and more attention from both academia and industry [46]. As shown in Figure 1, one can see that there were just a handful of publications each year before 2014. After that, the number of yearly publications has grown exponentially, reaching almost 500 publications in 2017. This exponential rate of increase continues, and the number of yearly publications exceeded 1000 in 2019. However, the growth of yearly publications slowed in 2020 because of the outbreak of COVID-19. Although the rapid spread of COVID-19 has led to a significant decrease in the publication of conference papers, NOMA remains a hot topic of research. As shown in Figure 1, the publications in journals are still increasing rapidly.

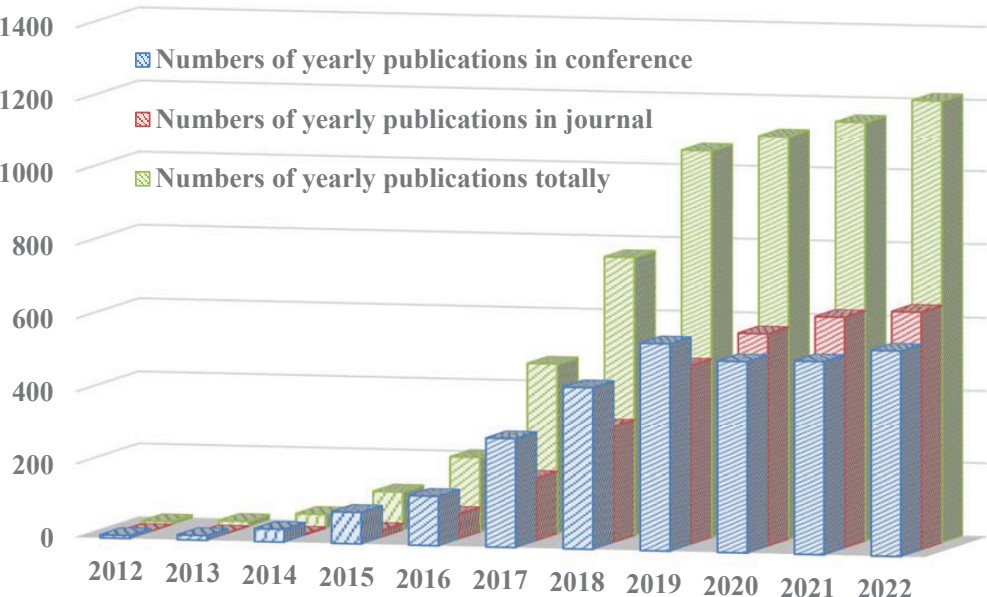

**Figure 1.** Cumulative results of yearly searches for publications using the acronym "NOMA" on the IEEE Xplore Digital Library over the past decade including conference papers, journal publications, and the total number of publications.

Among those publications, NOMA is mainly classified into two categories: power domain NOMA and code domain NOMA [47]. In this article, we focused on the former, and the term "NOMA" in the following denotes power domain NOMA. As is known to all, NOMA has its inherent advantages in terms of spectral efficiency, but some of its flaws, such as the single-resource-block-based design, the stringent requirements on users' channel qualities, the requirement of channel state information (CSI) at the transmitter, and the possible error propagation caused by performing successive interference cancellation (SIC), should be ameliorated or avoided. To this end, recent efforts to solve these flaws and the related research progresses are highlighted to provide some new insights into the potential future directions for researchers in this field [10,18,48–53]. In particular, unlike the existing single-resource-block-based NOMA structure, some novel multicarrier-based NOMA frameworks that combine transmit diversity and data multiplexing have been developed to achieve both high capacity and high reliability. The design principles and key features of such a framework are well discussed [50,51]. Besides, since deep learning approaches are powerful tools to provide solutions for networks where their system models are complex and difficult to describe with tractable mathematical expressions, the designs that join NOMA and deep learning are then discussed to seek high-performance data transmission for large-scale heterogeneous networks with low-cost signal processing [39–41]. In addition, some key challenges and future directions of the deep-learning-powered NOMA frameworks are also identified. Finally, the conclusions of this article are given.

## 2. Features of the Single-Resource-Block-Based NOMA Paradigm and Its Improvements

The current NOMA architecture is mainly a single-resource-block-based operation unit, in which its data exchange processes are performed on the same radio resources, and it was first conceived of to serve multiple user devices for the scenarios where their channel qualities are distinct. To be more specific, all users' intended symbols are multiplexed on the same single resource block at the transmitter side by utilizing superposition coding (SC), and then, each user demultiplexes its requested data at the receiver side via SIC. In practice, the SIC decoding order is naturally formed according to the difference in the users' channel conditions [46].

Single-resource-block-based structure design: For NOMA, the usage of SC to serve multiple users with the same radio resources enhances the system sum rate and user fairness well, but the intra-group interference, an inherent flaw caused by the scheduled near users, is inevitable and, thus, degrades the system reliability [10]. For example, the outage performance—a metric characterizing the reliability of systems—will be affected. Without loss of generality, assuming an $N$ users scheduled case and for the $i$-th user $U_i$, the outage performance is determined by the condition $\frac{\alpha_i}{\widetilde{\alpha}_i} \geq \gamma_i^{th}$, where $\alpha_i$ denotes the power allocation factor assigned for $U_i$'s decoding symbol $s_i$, $\widetilde{\alpha}_i = \sum_l \alpha_l$ represents the sum of the power allocation factors assigned for the other users who are treated as interference for decoding $s_i$ with $l \in \{1, \cdots, i-1\}$ or $l \in \{i+1, \cdots, N\}$, and $\gamma_i^{th}$ is the fixed threshold of the signal-to-interference-plus-noise ratio (SINR) (The threshold value $\gamma_i^{th}$ can be deduced from the predefined targeted rate, and the inherent flaw associated with the single-carrier NOMA paradigm can be solved by enhancing the received SINR through diversity-achieving transmission schemes.) for the SIC detector of $U_i$ to decode its intended symbol $s_i$ and other relevant symbols that must be first decoded. Noting that $U_i$ itself is the nearest user since SIC is applied, its intended symbol $s_i$ can be acquired usually after its relatively far users' symbols have been decoded. Due to the existence of inter-user interference, the degeneration of users' outage performance finally becomes more severe for cases where more than two users are co-scheduled to perform NOMA on this single-resource-block-based structure.

To avoid this issue, it is usually assumed that the SIC processor of $U_i$ can adjust its detection capability according to the current decoded symbol $s_j$ with $j \in \{i, \cdots, N\}$ or $j \in \{1, \cdots, i\}$, and $\gamma_i^{th}$ is written as $\gamma_{ij}^{th}$ for consistency. Here, $\gamma_j^{th}$ or $\gamma_{ij}^{th}$ denotes the dynamic predefined threshold of the SINR for decoding symbol $s_j$ at the processor of $U_i$. Typically, the threshold set for the far user is less than that of the near user considering that the far user suffers interference from the near one. In this sense, users' quality of service (QoS) requirements are taken into account. Therefore, the SIC processor's detection capability will be strengthened when decoding the far user's symbol and weakened while decoding the near one's data. For example, $\gamma_{ij}^{th}$ set for decoding the far user's data is far less than that of the near user's for the same SIC processing at $U_i$, as shown in [46]. Frankly, the detection capability of the SIC processor can be dynamically switched at the same user device, which sounds amazing, but may be impractical in the industrial context. The reason is that, for the processor of the near user, it requires decoding the symbols both of the far user and the near user itself. In this sense, the near user would not weaken the processing power when decoding its intended data in a practical application. On the contrary, it will take steps to strengthen the processing power to be comparable to that of the decoding of the far user's intended data, and thus, the predefined thresholds set for both the near and far users are similar. Finally, the phenomenon of the SIC decoder's detection capability being strengthened when decoding the far user's data and weakened when decoding the near user's data would not appear. Therefore, it is reasonable to assume that the processing capabilities for decoding the far and near users' intended data are equivalent for the processor in general, but such an assumption has rarely been considered in the existing single-resource-block-based NOMA structure.

Cognitive radio networking, a concept first introduced by Joseph Mitola [54], is another paradigm utilizing the QoS to improve system spectral efficiency, and from the viewpoint of cognitive radio technology, a user's device can be further classified as either a primary user device type or a secondary user device type. The IoT devices, for example, which are usually deployed in a massive machine-type communications system, wearable healthcare system, or autonomous unmanned system with the rigorous requirements of a low data rate, low latency, and high reliability, are usually treated as the primary user devices [55] and, thus, show a higher priority than their secondary counterparts. In the existing NOMA-based studies, for instance, the cell-edge user itself is equivalently viewed as the primary user, and its cell-center counterpart is, thus, regarded as the secondary user [10]. By introducing the aforementioned concepts, it is reasonable to assume that the

signal-detection capability of the primary user is more powerful than that of the secondary user, and thus, the primary user can be satisfied to decode its intended data for cases where strong interference is presented. But, one should bear in mind that the data rate required by the primary user is still low. A detailed analysis was presented in [48], where a novel user-scheduling strategy for which the decoding order can be dynamically adapted based on the users' CSI and QoS was established to realize NOMA transmission without outage probability error floors for the uplink communication scenario, thus achieving the goal of reliability improvement.

Besides, the Alamouti scheme is considered the most-efficient solution for the MISO broadcast channel. It not only has very low signaling and computational overhead for IoT devices, but also achieves an optimal balance between diversity and multiplexing. The authors in [56] were the first to attempt integrating the Alamouti encoding method with NOMA. This integration aimed to provide transmit diversity for cooperative communications systems and improve the received SINR. After that, there have been many studies on combining the Alamouti encoding method and NOMA to further enhance the reliability of data transmission in wireless communications systems [57–60]. Similar to the conventional Alamouti scheme, two time slots are needed for the Alamouti-NOMA scheme to transmit two superposition symbols $\left\{ s_{SC}^{l} \Big|_{l=1}^{2} \right\}$ to the user terminal. Specifically, the first transmitting antenna broadcasts $s_{SC}^{1}$ in the first time slot and $-s_{SC}^{2}{}^{*}$ in the second time slot, while the second transmitting antenna sends $s_{SC}^{2}$ and $s_{SC}^{1}{}^{*}$ during the first and the second time slots, respectively, with $(\cdot)^{*}$ denoting the conjugate operation.

However, the approach proposed in [48] is a partial-user-based scheduling scheme, in which only two users are selected to perform NOMA, and thus, the system spectral efficiency and massive connectivity cannot be fully exploited in practice. On the other hand, in the single-carrier NOMA structure, while far users may aim to improve their diversity order to compensate for the poor channel quality, near users may instead seek a higher multiplexing gain due to their favorable channel conditions. Such requirements, however, were not discussed in [56–60], which leads to the waste of the degrees of freedom and suggests that simply combining NOMA with the Alamouti encoding method may not be the optimal solution for improving system performance. To address these issues, multicarrier-based operation units were proposed to either boost the capacity by enjoying the multiplexing gain or improve the reliability by enjoying the diversity gain [10]. But, it is challenging for the single-cell edge user to pair with multiple near users to perform NOMA [61].

Stringent requirement on users' channel qualities: The superiority of NOMA over OMA is based on the prerequisite that users' channel qualities are distinct enough [11]. In other words, once the users' channel qualities are similar to each other, the advantages brought by performing NOMA will no longer exist compared with OMA. On the contrary, this will directly result in a higher system complexity when employing NOMA for those cases. Actually, the channel qualities of the scheduled users in the same resource block are sufficiently distinct, so it has difficulty satisfying these from the perspective of the network topology. Typically, users' distributions along the vertical direction will display varying channel conditions, namely they are located at different equal channel quality zones, and that means the network topologies with distinct channel gain are satisfied; conversely, users' distributions along the circumferential direction will display constant channel conditions, namely they are located at the same equal channel quality zone, and that means the network topologies with equal channel gain are satisfied. Therefore, there are four basic network topologies for the three-node co-scheduled case according to their locations (It has been reported that the optimal scheduling strategy is one in which only two or three nodes are co-scheduled in each resource block [62], and the three-node scheduling case was taken as a demonstrative example in this article.), that is all nodes' channel qualities are distinct, all nodes' channel qualities are similar, two nodes' channel qualities are similar and theirs are

better than that of the other one, and two nodes' channel qualities are similar and theirs are worse than that of the other one, as shown in Figure 2.

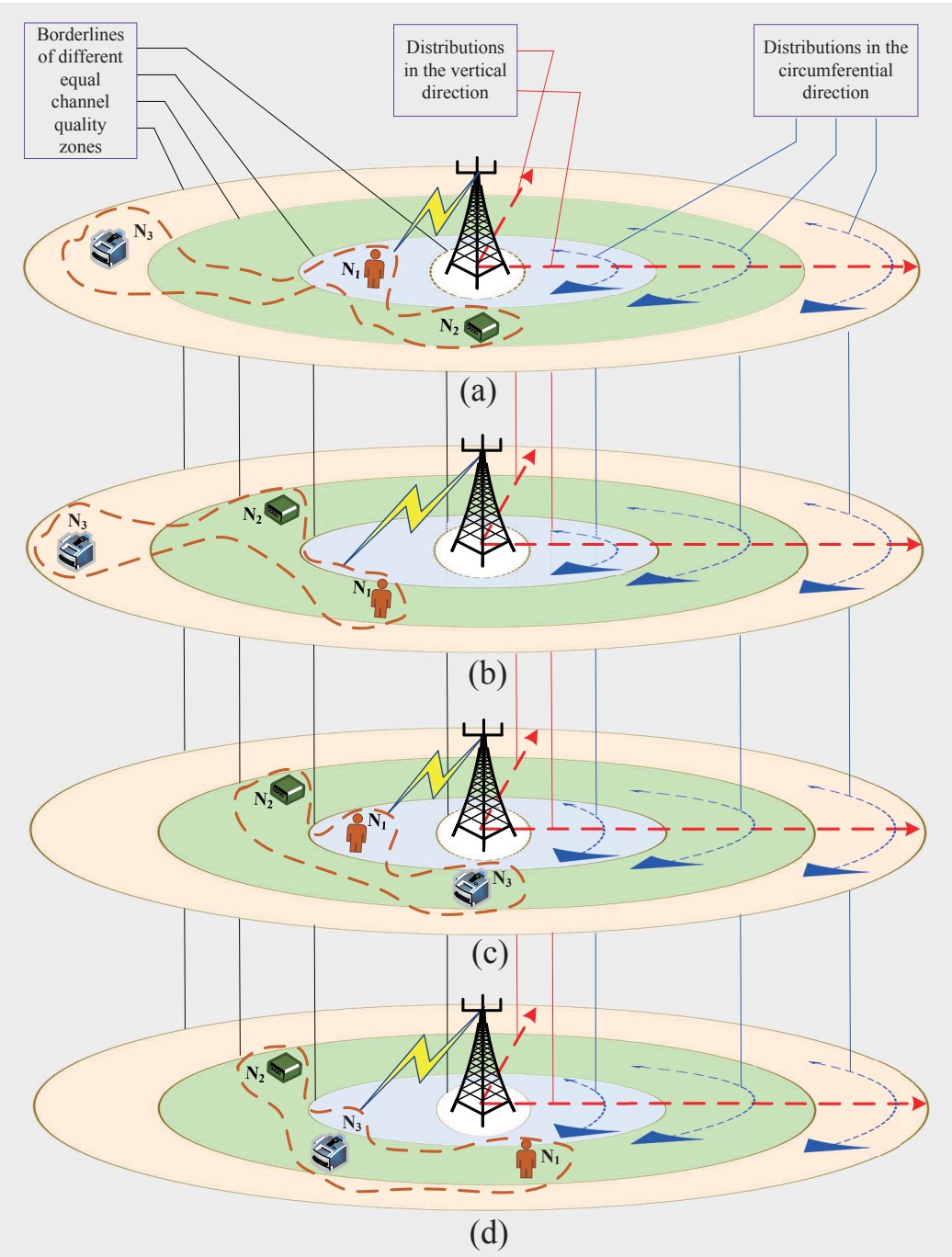

**Figure 2.** Basic network topologies with three co-scheduled nodes ($N_1$, $N_2$, and $N_3$) in a concentric model: (**a**) all nodes' channel qualities are distinct; (**b**) two nodes' channel qualities are similar and theirs are better than that of the other one; (**c**) two nodes' channel qualities are similar and theirs are worse than that of the other one; (**d**) all nodes' channel qualities are similar. The annular band along the direction of the arrow indicates that users have similar channel conditions.

As mentioned above, for the scenario where all nodes fall into equal channel qualities zones, as shown in Figure 2d, it is better to adopt OMA rather than NOMA or their hybrid to yield a reasonable performance gain. Therefore, this network topology is not discussed in this article for consistency. On the other hand, as is known to all, NOMA can yield

a reasonable performance gain over OMA under the condition that all scheduled users' channel qualities are distinct. Therefore, only the network topology in which the total users' locations fall into the vertical distributions is considered to perform NOMA, as illustrated in Figure 2a. Therefore, one will naturally ask how the pure NOMA structure (namely, the single-carrier framework) can relax or break this stringent channel quality requirement imposed on the scheduled users and perform effectively in those scenarios where some of their channel qualities are similar (For those scenarios where some users' channel qualities are not sufficiently distinct, it has been pointed out that the use of pure NOMA shows a better performance than pure OMA or their hybrid scheme [10]), as shown in Figure 2b,c.

Several attempts have been made in the literature to address this issue. For example, a bandwidth-division-based two-user pairing networking topology was first proposed to handle the case in which two far users with equal channel qualities are co-scheduled with a near user to perform NOMA [49]. Besides, the authors of [10] proposed a flexible bandwidth-division-based NOMA operation structure to improve the near users' reliability for those cases in which near users have identical channel qualities. To be more specific, the proposed NOMA operation structure was constructed by multiple virtual near–far user pairs, in which the far users are paired with each near user to perform NOMA at each sub-band. It was reported that the proposed pure NOMA greatly improves user fairness and the system's decoding complexity compared with the pure OMA or their hybrid scheme.

Perfect channel knowledge is needed: Different from OMA, for NOMA, the transmitter needs to know the perfect CSI to, on the one hand, select proper users/devices/equipment for grouping, clustering, or pairing scheduling such as the matching-theoretical algorithm or graph-theoretical approach and, on the other hand, develop an appropriate power budget scheme to perform SC for the scheduled users, and such a process carries the message of decoding, which will be passed to the receiver to form a specific SIC decoding order. Therefore, most of the existing works have assumed that the perfect CSI could be available at the transmitter. In reality, however, due to heavy feedback overhead, as well as harsh feedback delay constraints, it is a challenging task to obtain a perfect CSI at the transmitting side, especially for cases where massive MIMO and NOMA are applied jointly to strive for further performance improvement. To overcome this problem, it has been reported that, on the one hand, for cases where users have mobility, a class of low-feedback schemes has been developed. For example, a one-bit feedback-based users' channel quality order design has been proposed and compared with the order design with perfect channel knowledge; there was no performance loss for this low-feedback design scheme [51]. On the other hand, for cases where users have low mobility or a fixed location, it is a good choice to use the statistical CSI to form both SC and the decoding order at the transmitter and receiver, respectively, and a superior performance was clearly presented in [18].

Possible error propagation caused by employing SIC: Theoretically, the usage of SC at the transmitter enhances the system sum rate, user fairness, and scheduling flexibility, and the inter-user interference created by deploying SC can be eliminated by employing SIC at the receiver. This is the widely used structure design in NOMA for the spectral efficiency improvement [48,49,53]. However, a possible intrinsic weakness termed as error propagation could be introduced by the use of SIC, and this is an issue that needs to be taken seriously and well addressed [32]. The solution first conceived of by researchers is to assign more power budget to the far users, whose intended symbols usually need to be decoded first and many times [46]. Unfortunately, this approach will cause damage to system performance, especially for the cases where a large number of users are scheduled. Therefore, the authors in [48] proposed a reliable transmission scheme to remove the error floors caused by the error propagation, where the SIC decoding order can be dynamically adapted based on the users' CSI and QoS. Moreover, different from the above solutions proposed for the single-carrier-based NOMA systems, the generalized multicarrier framework [10,51] and the RIS-assisted structure, designed to enhance users' SINR to eliminate the error propagation [29,63], will be discussed in Sections 3 and 4 in detail, respectively.

On the other hand, users' information sources (ISs) in the aforementioned works were usually assumed to be independent. In fact, there are four different cases according to the correlation of the ISs in the multiple-access channel, including the other cases where users' ISs are correlated [53], as shown in Table 1. Therefore, for the cases where the users' ISs are correlated, the SIC technique would still be needed; in other words: Would the SIC technique would still be the best performance booster? The answer is no, and this fact was revealed in [53]. To be more specific, for cases where the correlation of users' ISs is greater than 0.8, the author in [53] pointed out that there is no capacity loss with the proposed non-SIC NOMA scheme. The non-SIC NOMA scheme here denotes that the near user can directly decode its intended data by treating the far user's data as noise, thus greatly reducing its decoding complexity compared to the existing SIC-based NOMA scheme and totally eliminating the error propagation problem [10,18,47–49,52].

**Table 1.** The correlation of information sources for a two-user case.

| | | Relationship of Users' Coded ISs (CISs) | |
| | | Independent | Correlated |
| --- | --- | --- | --- |
| Relationship of Users' Original ISs (OISs) | Independent | $\rho_{i,j}^{OISs} = 0, \rho_{i,j}^{CISs} = 0$ | $\rho_{i,j}^{OISs} = 0, \rho_{i,j}^{CISs} \neq 0$ |
| | Correlated | $\rho_{i,j}^{OISs} \neq 0, \rho_{i,j}^{CISs} = 0$ | $\rho_{i,j}^{OISs} \neq 0, \rho_{i,j}^{CISs} \neq 0$ |

## 3. Evolutions towards the Generalized Multiple-Resource-Block-Based NOMA Frameworks

It is a common myth that the use of SIC is what will lead to the problem of error propagation in NOMA. In fact, such a belief is exactly incorrect, and from the perspective of information theory, this problem can be addressed by using a multicarrier-based NOMA design to improve the transmitted data's SINR. The concept of a multicarrier-based NOMA design was first proposed by the authors in [10,50] to handle the cases where users' channel qualities (the far users' and the near users') are similar or not distinct enough, as illustrated in Figure 2b,c. The core idea behind these solutions is that the single-resource block (this usually means the frequency band resource) is divided into multiple sub-blocks/sub-bands and all users are accommodated in each sub-block/sub-band to perform NOMA [51,64]. Moreover, only far users employ repetition-based transmission across all sub-bands to acquire diversity gain by performing maximum ratio combining (MRC) at the receiver. Finally, a combination of MRC and SIC is integrated in each sub-block/sub-band to take advantage of the parallel processing for data decoding. The key difference between the works [10,50] is that the former focuses on improving the reliability of data transmission via enhancing the diversity gain, while the latter aims at boosting the achievable capacity by harvesting the multiplexing gain. For those network topologies in which users' channel qualities are distinct, on the other hand, as shown in Figure 3a, a multicarrier-based NOMA networking design, which can improve the cell-edge users' SINR significantly, is proposed to tackle problems caused by SIC, as described in [51]. Unlike the works [10,50], the scheme presented in [51] is a simple and subtle design, in which the diversity–multiplexing gain can be harvested for NOMA to achieve both the high performance of ultra-reliability and high-capacity. More specifically, repetition-based coding is employed for the far users at the transmitter, and the maximum ratio combination (MRC) is integrated with SIC to jointly decode their intended symbols at the receiver. By doing so, on the one hand, NOMA's merit is that the spectral efficiency dominated by near users has not been weakened; on the other hand, NOMA's drawback is that the reliability limited by the far users can be broken. However, the correlation between the repetition-based coding implementation and the frequency band resource scheduling has not been well discussed, which is a fundamental tradeoff related to diversity–multiplexing that one should be concerned about in the above multicarrier NOMA design.

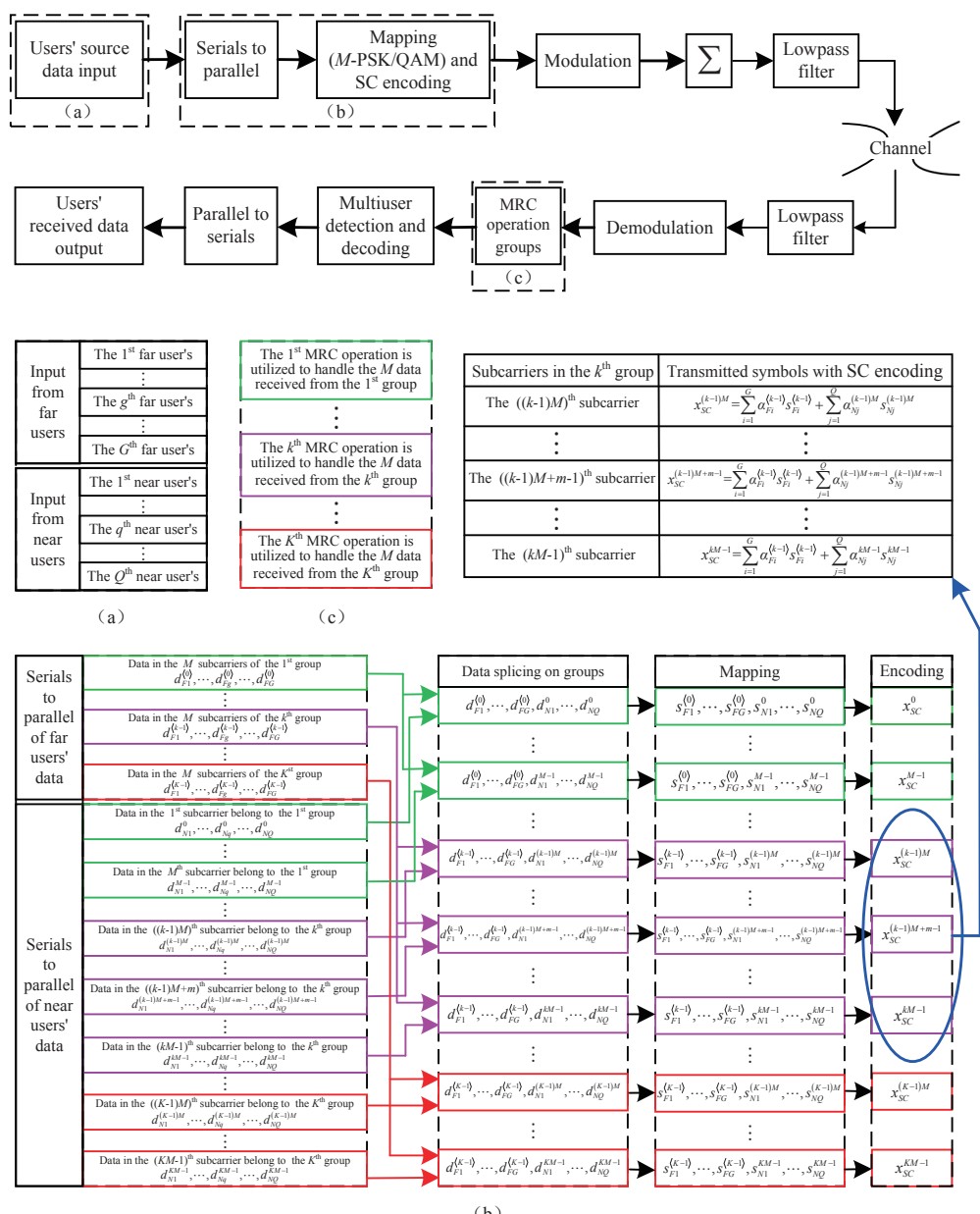

**Figure 3.** Baseband model of the MRC-enabled multicarrier NOMA communications system: (**a**) details of the inputs of users' source data; (**b**) processes of the data's transformation from serial to parallel, mapping, and encoding; (**c**) specifications of the MRC operation groups.

To provide inspiration and guidance for readers on how to understand this tradeoff, in this section, we elaborate on a more-generalized MRC-enabled multicarrier NOMA framework, as shown in Figure 3, in which $N$ subcarriers with $N = KM$ are shared by users with $G$ far ones and $Q$ near ones. $K$ and $M$ here denote that the $N$ subcarriers are divided into $K$ groups and each group has $M$ subcarriers, and $k \in \{1, \cdots, K\}$ and $m \in \{1, \cdots, M\}$ denote the subcarriers in the $k$-th group and the $m$-th subcarrier in the $k$-th group, respectively. Here, the repeated transmission is applied to the far users for data exchange, and their required dataset, $\left\{ d_{Fg}^{\langle k-1 \rangle} \Big|_{g=1}^{G} \right\}$, will be transmitted repeatedly in every $M$ subcarrier of the $k$-th group to harvest the diversity gain with the MRC-operation-group-based receiver, as shown in Figure 3c; thus, only $K$ data will be sent in each time slot. Note that the letters $d$ and $F$ denote the data required by the far users, and $g$ and $\langle k-1 \rangle$ denote the $g$-th far user's required data $d_{Fg}^{\langle k-1 \rangle}$ transmitted in each of the subcarriers

of the $k$-th group, with $g \in \{1, \cdots, G\}$ and $k - 1 = \left\lfloor \frac{(k=1)M+m=1}{M} \right\rfloor$. On the other hand, none of the near users apply the repeated transmission scheme, and their required data will be transmitted in each subcarrier; for example, the transmitted dataset in the $m$-th subcarriers of the $k$-th group is $\left\{ d_{N_q}^{(k-1)M+m-1} \Big|_{q=1}^Q \right\}$, and thus, there are $KM$ different data that are finally sent in each time slot. Similarly, the letters $d$ and $N$ denote the data required by the near users, and $q$ and $(k-1)M + m - 1$ denote the $q$-th near user's required data $d_{N_q}^{(k-1)M+m-1}$ transmitted in the $((k-1)M + m)$-th subcarrier of the $k$-th group, with $q \in \{1, \cdots, Q\}$. Finally, the symbol $s_{SC}^{(k-1)M+m-1}$ transmitted in the $m$-th subcarrier of the $k$-th group is shown in Figure 3, encircled by an oval-shaped mark, and its specific form is formulated in the corresponding part in an enlarged manner, with $\alpha_{F_i}^{\langle k-1 \rangle}$ and $\alpha_{N_j}^{(k-1)M+m-1}$ denoting the power allocation factor of the i-th far user in the $k$-th group and the power allocation factor of the $j$-th near user in the $(k-1)M + m$-th subcarrier of the $k$-th group, respectively.

By varying parameters $G$, $Q$, $K$, and $M$, the proposed NOMA-based generalized multicarrier framework can usually be degraded as the existing NOMA structures. For example, it is degraded as the classic two-user pairing single-carrier NOMA system [46] with $KM = 1$ or it is degraded as the existing multicarrier frameworks such as the conventional OFDM-based NOMA system [19,20] and the multiple-pairwise-based NOMA system [10] when $M = K = 1$, respectively. Besides, letting $G = 2$, $Q = 1$, $K = 2$, and $M = 1$ and with the assumptions that the near user can enjoy both subcarriers and the two far users have the same channel gains, the proposed framework can further be degraded as the system presented in [50]. The results are summarized in Table 2, where the diversity-multiplexing gain is fully characterized by $K$ or $M$. Specifically, increasing $K$ (meaning that a decrease in $M$) results in higher multiplexing gain for the near users and lower diversity gain for the far users, and vice versa [51]. To demonstrate the superiority of the multicarrier NOMA scheme, finally, the performance comparison is presented in Figure 4, highlighting its advantages over other schemes such as the OMA scheme and the single-carrier NOMA scheme over Rayleigh flat-fading channels (The formulas that allow us to obtain the results of the sum rate and the outage probability were given and proven by the authors in [51]. For simplicity, they are omitted here).We use the terms SC-NOMA and MC-NOMA to denote the single-carrier NOMA scheme and the multicarrier NOMA scheme, respectively. Their comparison results are shown in Figure 4, in which the simulation parameters are set as: $G = 1$, $Q = 2$, $K = 1$, $M = 2$, the channel variances of $U_1$, $U_2$, and $U_3$ are $\sigma_1^2 = 10$, $\sigma_2^2 = 6$, and $\sigma_3^2 = 2$, respectively, the targeted rate for each user is $R_i^{th} = 1.5$ bps/Hz, and the total bandwidth is $B = 1$. Besides, the power allocation factors assigned for $U_1$, $U_2$, and $U_3$ with NOMA are $\alpha_1 = 0.6$, $\alpha_2 = 0.3$, and $\alpha_3 = 0.1$, respectively, while the ones assigned for users with OMA is equal. Since the average sum rate decreases as $M$ increases, one can see that the average sum rates of MC-NOMA are worse than those of SC-NOMA. But, fortunately, the loss for MC-NOMA is negligible since its sum rate is mainly determined by the near user, whose performance is not affected by changes in $M$. Besides, the average sum rates of MC-NOMA are greatly superior to those of OMA in the whole SNR region. On the other hand, due to the diversity gain that is able to be achieved, the outage probability of MC-NOMA is superior to that of SC-NOMA, especially for those cases where the users' rate thresholds are relatively high, as shown in Figure 4b.

**Table 2.** The proposed generalized multicarrier framework is degraded in the existing NOMA schemes with varying parameters.

| Conditions | | Existing Works |
|---|---|---|
| **Parameter Setting** | **Channel Setting** | |
| $G = 1, Q = 1, KM = 1$ | Channel qualities are distinct | [46,48] |
| $M = 1$ | Channel qualities are distinct | [39,40] |
| $K = 1$ | Near users' channel qualities are distinct | [10] |
| $G = 2, Q = 1, K = 2, M = 1$ | Far users' channel qualities are distinct | [50] |

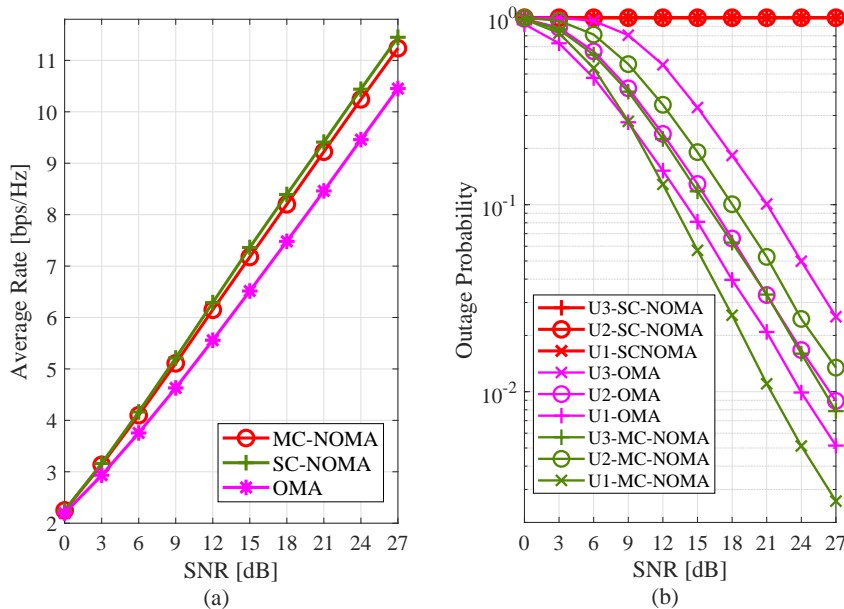

(a)                                                    (b)

**Figure 4.** Performance comparisons among the OMA scheme, the single-carrier NOMA scheme, and the multicarrier NOMA scheme with three users scheduled: (**a**) ergodic sum rates versus the transmit signal-to-noise ratio (SNR); (**b**) outage probability versus the transmit SNR.

## 4. Developments towards RIS-Assisted NOMA Systems

Due to its strong capability of reconfiguring the reflected signal propagations intelligently, the RIS has been viewed as a powerful solution to enhance the performance of NOMA [27,28]. It has been pointed out that the RIS can offer additional channel paths to create stronger combined channels with noticeable strength differences. Additionally, it can artificially realign the combined channels of users to achieve NOMA gains for challenging scenarios where the channel qualities of different users are similar [30].

According to the basic network topologies, the RIS-assisted NOMA systems, which usually comprise a source node, BS, the intelligent reflective surface node, RIS, and multi-user nodes, U, can be simply divided into four categories, namely the uplink/downlink single-RIS union-assisted multi-user scenario, the uplink/downlink single-RIS partition-assisted multi-user scenario, the uplink/downlink multi-RIS union-assisted multi-user scenario, and the uplink/downlink multi-RIS partition-assisted multi-user scenario, as shown in Figure 5a–d. Without loss of generality and for simplicity, the downlink single-RIS-assisted NOMA network was taken as an example to analyze how to exploit interference cancellation to enhance the SINR in this paper. We, more specifically, considered a downlink single-RIS-assisted NOMA network, in which $N$ multiple users $\left\{ U_n |_{n=1}^N \right\}$ are served by a single BS and an RIS consisting of $V$ elements, as shown in Figure 5a,b. Let $\mathbf{f} \in \mathbb{C}^{V \times 1}$ with $\mathbf{f}^T = [h_1, \cdots, h_v, \cdots, h_V]$, $\mathbf{g_n} \in \mathbb{C}^{V \times 1}$ with $\mathbf{g_n}^T = [h_{1,n}, \cdots, h_{v,n}, \cdots, h_{V,n}]$,

and $h_n$ denote the channel coefficients of the BS-to-RIS, RIS-to-$U_n$, and BS-to-$U_n$, respectively. Besides, letting $\boldsymbol{\Theta} \in {}^{V \times V}$ denotes that, for the RIS's diagonal phase shift matrix with $\theta_v \in [0, 2\pi)$ denoting the phase shift of the $v$-th reflecting element, we have $\boldsymbol{\Theta} = diag\{e^{j\theta_1}, \cdots, e^{j\theta_v}, \cdots, e^{j\theta_V}\}$. For the cases without the assistance of the RIS, we further assumed that the end-to-end channel gains $\left\{ |h_n|^2 \Big|_{n=1}^N \right\}$ have a fixed user order for each channel realization. Without loss of generality, this fixed user order was assumed to be $\left\langle \pi_{w/o} : |h_1|^2 \leq \cdots \leq |h_n|^2 \leq \cdots \leq |h_N|^2 \right\rangle$, and the SINR of $U_n$ to decode its intended symbol in that case can be formulated as $SINR_{w/o}^n = \frac{\rho |h_n|^2 \alpha_n}{\rho |h_n|^2 \sum_{i=n+1}^N \alpha_i + 1}$ with $\rho$ denoting the SNR [46,50].

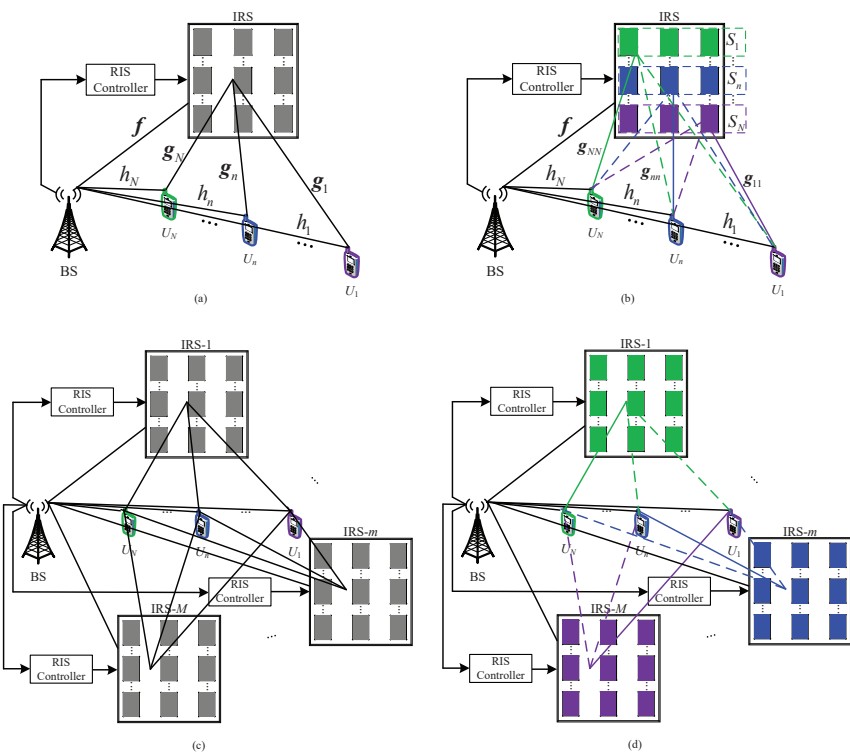

**Figure 5.** Network topologies of RIS-assisted NOMA systems with channels ordered: (**a**) uplink/downlink single-RIS union-assisted multi-user scenario; (**b**) uplink/downlink single-RIS partition-assisted multi-user scenario; (**c**) uplink/downlink multi-RIS union-assisted multi-user scenario; (**d**) uplink/downlink multi-RIS partition-assisted multi-user scenario. The solid line denotes the information link, and the dashed line denotes the residual interference.

The downlink RIS union-assisted multi-user system: All the reflecting elements adjust their phase shifts with a unified control/processing strategy, termed as the RIS union-assisted design. In this case, $U_n$'s equivalent end-to-end channel coefficients, denoted by $\tilde{h}_n$, can be formulated as $\tilde{h}_n = h_n + \mathbf{g_n}^T \boldsymbol{\Theta} \mathbf{f}$. This means that, by intelligently tuning $\boldsymbol{\Theta}$, RIS-assisted NOMA can effectively enhance the system design by introducing more degrees of freedom (DoFs). In simpler terms, this technology can not only create a considerable disparity in the channel gains experienced by different users, but also allow the customization of their effective channel gains to meet their QoS requirements [27]. More specifically, by adjusting the phase shift of the reflecting element with unified control/processing in $\boldsymbol{\Theta}$, for example, the optimal user order $\pi_w$ may be any one of all the $V!$ different user orders with $\left\langle \pi_w : |\tilde{h}_N|^2 \leq \cdots \leq |\tilde{h}_n|^2 \leq \cdots \leq |\tilde{h}_1|^2 \right\rangle$, which is totally different from the fixed order $\pi_{w/o}$ given in the NOMA networks without the assistance of the RIS. From this perspective, we can effectively reduce the computational complexity of successive interference cancella-

tion (SIC) by flexibly adjusting users' decoding order based on the actual needs. On the other hand, by adjusting the phase shift values in $\mathbf{\Theta}$ to make the direct signal (originated from the BS-to-$U_n$ link) and the reflected signal (originated from the BS-to-RIS-to-$U_n$ link) have the same phases at receiver $U_n$, we will have that $\left|h_n + \mathbf{g_n}^T\mathbf{\Theta f}\right|^2 \geq |h_n|^2$ holds. The SINR of $U_n$, denoted by $SINR_w^n = \frac{\rho\left|h_n + \mathbf{g_n}^T\mathbf{\Theta f}\right|^2 \alpha_n}{\rho\left|h_n + \mathbf{g_n}^T\mathbf{\Theta f}\right|^2 \sum_{i=n+1}^N \alpha_i + 1}$, will be improved significantly compared with $SINR_{w/o}^n$. Finally, the capacity, reliability, and error propagation of the RIS-assisted NOMA system will be significantly enhanced.

The downlink RIS partition-assisted multi-user system: The reflecting elements' phase-adjusting strategy with the divide and rule strategy is more effective than that of the unified control/processing in reality, especially for the cases where the locations of the users are clustered in a distributed manner around the RIS or users are randomly located in the vicinity of the RIS, and the superiorities were shown in [31,32]. This divide and rule strategy is termed as the RIS partition-assisted design. In this case, all the *V* elements are partitioned into *N* subsurfaces, and the *n*-th subsurface, denoted by $S_n$, consists of $V_n$ elements, as shown in Figure 5b. Therefore, when compared to the RIS union-assisted design, the RIS partition-assisted design poses greater challenges in accurately adjusting the phase shifts of each subsurface. This is because, in the partition-assisted design, all subsurfaces' phase adjustments must be coordinated with each other to achieve optimal interference reduction. As a result, this may lead to significant computational overhead for each channel realization.

## 5. DL-Enabled NOMA Frameworks

In general, conventional data-processing methods in communications systems are usually built on tractable mathematical models, in which the ideal hypotheses, for example linear, stationary, and so on, are assumed. However, most of the practical systems are non-linear and/or time varying, and thus, they cannot be fully captured by such algorithms. The reasons why the conventional data-processing methods could not be the optimal choice for communications systems are typically twofold: the first is the current modular systems' design, whose signal-processing blocks such as the coding, modulation, and equalization only carry out their individual tasks, making it difficult for them to find the optimal solution, and second is that there is not always an effective treatment for those complex systems such as large-scale heterogeneous wireless networks, which limits their practical implementation.

On the other hand, AI or, more precisely, DL is a powerful tool to provide better performance for communications systems, especially for the RIS-assisted NOMA networks, where the system models are complex and difficult to describe with tractable phase adjusting and wireless resource management [33,36]. In other words, the powerful information-processing capabilities are the most-attractive advantage of DL [37], which makes it effectively handle the problems related to communications systems that cannot be solved mathematically. Therefore, DL for wireless communications is a hot spot in current research, and it has been widely applied to assist the design of communications systems, including aspects of the system modeling such as channel estimation/equalization, high-performance beamforming construction, constellation/codebook design, signal/symbol processing, multi-user detection, and so on, as well as the resource management, such as spectrum resource allocation, energy resource control, cloud resource optimization, opportunistic user scheduling, computing resource on- and off-loading, and so on.

In general, DL-enabled frameworks can be categorized into two major groups, that is data-driven and model-driven architectures (The pseudo-code and equations of the considered deep learning algorithm are the same as those of the existing approaches whose pseudo-code and equations have been well formulated. Therefore, for the sake of simplicity, we omitted the pseudo-code and equations here).A data-driven architecture shows less effort for system construction, but more effort for training data to make the system stabilize and converge. Conversely, a model-driven architecture can relieve the load

on the training data to promote learning efficiency with the domain knowledge (also termed as prior information) of the system; but, accurate domain knowledge is usually difficult to obtain in a real environment. Fortunately, such knowledge has been fully explored and exploited over several decades for wireless communications systems. In a word, by integrating NOMA and the deep learning approach, there will be a significant performance improvement on the above-mentioned tasks. In the next section, the deep-learning-enabled NOMA frameworks are discussed in detail.

### 5.1. Deep Learning for System Modeling

The model-driven architecture, as one of the typical paradigms of deep learning, is a powerful tool for optimization in physical layer communications. Unlike the conventional optimization methods, deep-learning-based data processing can be flexibly operated in either a point-to-point (local) or an end-to-end (global) manner to find the optimal solution for system modeling. In a local optimization case, for example, an optimal receiver is designed for symbol detection with the assistance of a model-driven architecture [39], as illustrated in Figure 6a (the baseband transceiver design of [39] is not shown in Figure 6a because it has the same structure as that presented in Figure 3 with $M = 1$), in which the functions of channel estimation/equalization and symbol demodulation are fulfilled simultaneously. More specifically, $L$ users' intended data are scheduled in each OFDM subcarrier to perform NOMA, and the quadrature phase-shift keying (QPSK) modulation is chosen to map their intended bits to symbols at the transmitter. At the receiver, the received signal will be first demodulated and, then, input into the mentioned deep-learning-enabled data-processing unit—a structure with seven layers, namely the reshape layer, convolutional layer, batch normalization layer, Relu activation function layer, flatten layer, long short-term memory (LSTM) layer, and fully connected (FC) layer with the softmax activation function, as shown in Figure 6a.

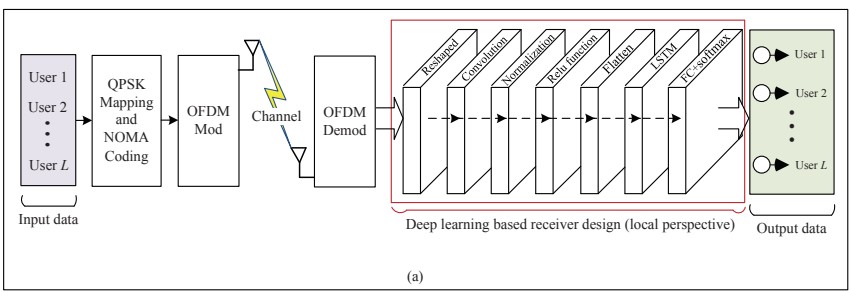

(a)

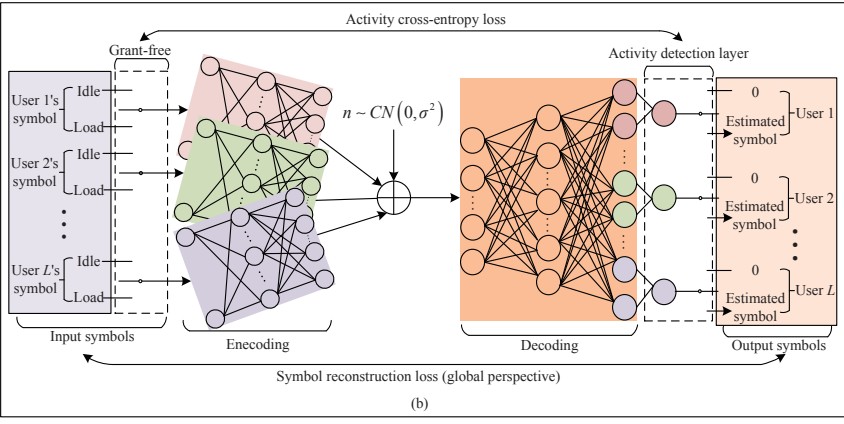

(b)

**Figure 6.** Illustrations of the deep-learning-enabled frameworks for NOMA: (**a**) a local optimization case with the assistance of a model-driven architecture; (**b**) a global optimization case with the assistance of a data-driven architecture.

The key difference between the conventional method and the deep learning approach is that the deep learning design is proposed to simultaneously fulfil the functions of channel estimation, equalization, and demodulation, which brings a gain coming from such a whole-system optimization perspective—something not possible for traditional methods. Besides, compared with the conventional methods, the gain achieved by the deep learning approach is higher in general, especially for the cases where the number of usersis small and the near user's intended symbol is decoded after the far user's one. For example, the gaps of the far and near users between the conventional MMSE-SIC detection algorithm and the deep learning approach at a symbol error rate (SER) value of $10^{-2}$ can be up to 1 dB and 5 dB [39], respectively. Furthermore, the deep learning approach has less computational complexity as compared with the conventional data-processing methods such as the above-mentioned MMSE-SIC detection algorithm, in which the intensive computational requirement for the matrix inversion operation is usually inevitable.

Typically, the model-driven architecture is suitable for the optimization of the local system, while the data-driven architecture is used to construct a global system. The authors in [41] proposed a pure data-driven architecture, termed as the fully connected deep neural network (FC-DNN), to find the global optimal solution to improve the reliability of a grant-free NOMA network, as shown in Figure 6b. Compared with the existing NOMA scheme, their results showed that there was about a 5 dB gain in the SER. More details can be found in [41].

### 5.2. Deep Learning for Resource Management

Resource management, such as user scheduling, channel assignment, and/or power allocation, is a key issue that should be taken into account in performing NOMA. However, the traditional resource-management methods have considerable complexity, especially for large-scale heterogeneous networks, where the optimum designs/solutions are not known or not available, thus leading to practical limitations. The deep learning approach can play an essential role in addressing the challenges of resource management in such networks. The key advantage of such an approach is that the deep-learning-enabled framework, as stated above, can flexibly carry out the design of optimizing performance in either a local or a global manner, and moreover, it will not introduce any additional transmission overhead or latency compared to the conventional methods. In [40], a deep-learning-aided low-complexity real-time resource allocation algorithm, termed as PAUS, was proposed. In this case, the resource management is performed in two stages, that is the power allocation stage and the user scheduling stage. In the first stage, the power allocation is implemented to optimize the system sum rate by using the interior point method (IPM)—a way of processing data that can accelerate the convergence and calculation speed of the training of deep neural networks; in the second stage, a dynamic networking algorithm is carried out to perform the user scheduling globally, and thus, the system sum rate can be further improved. It was shown that the achievable sum rate of the NOMA system can be improved significantly by employing deep learning with the proposed PAUS algorithm.

### 5.3. Challenges and Future Directions

As discussed above, deep learning and NOMA comprise an effective means of transmission to accomplish data exchange, and by doing so, the system spectral efficiency can be improved significantly. However, this does not mean that deep-learning-based NOMA structure designs can be completely transplanted to the 5G and beyond networks. As is known to all, unified data networking and processing across large-scale heterogeneous networks comprise the direction for the mainstream development of future communication networks. The devices/user equipment belonging to subnetwork systems such as the IoT, wireless sensors, massive machine-type communications (mMTC), and vehicle-to-everything (V2X) in the heterogeneous network usually have limited data-processing capabilities, but at the same time, they have very high expectations for the need to build high-reliability, low-complexity, and low-delay methods during signal processing. Al-

though deep learning is an effective way to address the problems of network systems, it needs tonnes of data and many attempts to succeed on very specific problems, and furthermore, it has difficulty generalizing its knowledge to very different tasks compared to those it has trained upon. For example, the knowledge learned with different encodings/decodings may vary greatly. In this sense, this is a really tough task for devices whose data-processing capabilities are limited, thus causing unacceptable complexity and delay for systems. The ways to circumvent the problem are mainly threefold: algorithm design, big data utilization, and computing power configuration. Therefore, on the one hand, considering the variability of the dynamical topologies of heterogeneous networks, deep-learning-enabled unified frameworks for NOMA should be primarily built to cut the computational complexity and data processing delay from an end-to-end perspective [20]; on the other hand, new deep-learning-based data-processing algorithms, which can further reduce the complexity and delay at the expense of marginal performance degradation from a point-to-point perspective, need to be developed urgently. A specific example of using a step-by-step resource management strategy to reduce the complexity of real-time resource management was discussed in an earlier subsection [19]. It is highly expected that the deep-learning-enabled frameworks for NOMA will be able to be carefully designed to meet the demands of various application scenarios in the 5G and beyond networks.

## 6. Conclusions

In this article, recent efforts to remove the flaws of the single-resource-block-based NOMA structure were fully revealed, and some significant research progresses have been made in the aspects of the system design, near–far effect utilization, channel knowledge management, and error propagation control. Besides, different from the conventional single-carrier structure, a generalized multicarrier-based NOMA framework has been conceived of and developed for achieving both high reliability and high spectral efficiency. In such a framework, the design of encoding and decoding, the complexity of the hardware implementation, and the flexibility of the system extension have been clearly stated. Furthermore, RIS-assisted NOMA networks have been developed to improve the receivers' SINR through the intelligent reconfiguration of the reflected signal propagations. The system designs with RIS union assistance and RIS partition assistance were both discussed in detail. Finally, the designs that combine NOMA/RIS-NOMA and deep learning to accomplish high-efficiency data transmission at a low cost were comprehensively discussed from both the local and global optimization perspectives. Finally, some key trends and future directions for the design of the deep-learning-based NOMA frameworks were also highlighted.

**Author Contributions:** Conceptualization, D.W., R.H. and G.D.; methodology, J.Y. and Z.H.; software, Z.H. and D.W.; validation, D.W., T.H. and J.Y.; formal analysis, D.W., M.W., T.H. and R.H.; investigation, D.W., G.D., L.C., J.L. and T.F.; data curation, Z.H. and T.H.; writing—original draft preparation, D.W. and R.H.; writing—review and editing, D.W., R.H. and G.D.; supervision, D.W., M.W., and F.J. All authors have read and agreed to the published version of the manuscript.

**Funding:** The work of Ronglan Huang was supported in part by the Natural Science Foundation of Guangxi Province under Grant 2020GXNSFBA297037 and in part by the Key Scientific Research Funds of Wuzhou University under Grant 2020B001. The work of Dehuan Wan was supported in part by the National Natural Science Foundation of China under Grant 61971149, in part by the Guangdong Basic and Applied Basic Research Foundation under Grant 2021A1515011657, and in part by the Special Foundation in Key Fields for Colleges and Universities of Guangdong Province (New Generation of Information Technology) under Grant 2020ZDZX3025.

**Data Availability Statement:** The data of global machine-to-machine connections involved in this paper can find in the website: https://www.marketresearchfuture.com/reports/machine-2-machine-connections-market-3818.

**Conflicts of Interest:** The authors declare no conflict of interest.

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
