# Peer review of "Towards Flawless Designs: Recent Progresses in Non-Orthogonal Multiple Access Technology"

_electronics, doi:10.3390/electronics12224577_

Round 1

Reviewer 1 Report

Comments and Suggestions for Authors

The paper presents a brief overview of power-domain NOMA and highlights some future research directions.

My comments are:

- The scope of the paper has to be specific as a large volume of research about NOMA and its integration with the other schemes is there. For example, several survey papers, which are not cited here, were written about NOMA or even about a specific aspect of NOMA with hundreds of references!

- In relation to the comment above, how about MIMO-NOMA, holographic MIMO-NOMA, NOMA and edge computing,,,,,,,

- The section about NOMA and deep learning is too brief and similarly the future research section. 

- The relevance of Fig. 2 in not clear and the results in Fig. 4 are not explained. As an example, why should SC-NOMA and MC-NOMA have similar sum rates? 

- Any simulations for RIS-assisted NOMA schemes? 

Comments on the Quality of English Language

Several typos and grammar issues are there, and they have to be fixed by careful proofreading. 

Author Response

Dear Reviewers,

Thank you very much for reviewing our paper. We appreciate your insightful comments, which helped us significantly improve the overall technical as well as presentation quality of this submission.

We hope the anonymous Reviewer will find that the review comments are sufficiently addressed and the current paper is acceptable for publication in Electronics.

With our sincere thanks once again.

Best regards,

All authors

Reviewer 2 Report

Comments and Suggestions for Authors

This paper provides a comprehensive study, comparison, and classification of the current advanced NOMA schemes from the perspectives of single–carrier systems, multi–carrier systems,  reconfigurable intelligent surface (RIS) assisted systems, and deep learning (DL) assisted systems.  Specifically, system implementation issues such as the transition from single-carrier to multi–carrier  systems, the relaxation of distinct channel gains, the consideration of imperfect channel knowledge,  and the mitigation of error–propagation/intra-group interference are involved. To begin with, we  present an overview of the state–of–the–art developments related to the advanced design of single-carrier NOMA system. Subsequently, a generalized multi–carrier NOMA framework that provides the diversity–multiplexing gain by enhancing users’ Signal–to–Interference plus Noise Ratio (SINR)  is proposed for a better system performance.

the article is of good scientific quality, but there are several points to bear in mind

1-The abstract is too long and it is difficult to see the main contribution of the article. 

2- Authors should provide a good motivation for their articles.

3- I found that the article lacks a model and it is difficult to interpret the results obtained.

4- Figure 1 is unclear and a reference should be added. 

5- Several references are added without you detailing their scientific limits or comparing them to your work.

6-please give more details on your mathematical model and briefly explain your model in figure 3.

7-add the formulas that allow you to obtain the results of the flow rate and the outage probability.

8- add a table of simulation values. I find that your method does not give a large difference compared with the classic method (with which you compare yourself). a comparison with an article in terms of results is necessary.

9- What type of channel is used in your model?

10-the SINR equation must be detailed and why do you neglect interference.

11-The proposed deep learning algorithm is poorly explained. Please add pseudo-code and the equation used during the simulation for better visibility.

12- What are the limits of your approach and the competational time of your algorithm?

13- Why are no results added to this section?

please add this paper to your research

Singh D, Ouamri MA, Muthanna MSA, Adam ABM, Muthanna A, Koucheryavy A, El-Latif AAA. A Generalized Approach on Outage Performance Analysis of Dual-Hop Decode and Forward Relaying for 5G and beyond Scenarios. Sustainability. 2022; 14(19):12870. https://doi.org/10.3390/su141912870.

D. Alkama, S. Zenadji, M. A. Ouamri, A. Khireddine and M. Azni, "Performance of Resource Allocation for Downlink Non-Orthogonal Multiple Access Systems in Tri-Sectorial Cell," 2022 IEEE International Conference on Electrical Sciences and Technologies in Maghreb (CISTEM), Tunis, Tunisia, 2022, pp. 1-6.

Comments on the Quality of English Language

English improved required

Author Response

Dear Reviewers,

Thank you very much for reviewing our paper. We appreciate your insightful comments, which helped us significantly improve the overall technical as well as presentation quality of this submission.

We hope the anonymous Reviewers will find that the review comments are sufficiently addressed and the current paper is acceptable for publication in Electronics.

With our sincere thanks once again.

Best regards,

All authors

Round 2

Reviewer 1 Report

Comments and Suggestions for Authors

The paper is improved now. However, 

- Make sure that all the simulations parameters used for Fig. 4 are stated in the paper (text or in a table), along with ref [51]. 

- It is more common to write "intra-cluster" rather than "intra-group" interference. 

- Some typos are there such as "by thenear user" on page 11,...

Also, "n, the year of 2021 would receive a record that about 28 billion smart 31 devices are connected across the world, and meanwhile, more than 15 billion machine-to-machine 32 (M2M) communication devices are deployed" is outdated information!

- Is Fig. 4-b sufficiently discussed in the text? 

Comments on the Quality of English Language

The English is clear; hwoever, the paper has to be proofread. 

Author Response

Dear Reviewers,

Thank you very much for reviewing our paper. We appreciate your insightful comments, which helped us significantly improve the overall technical as well as presentation quality of this submission.

Specific revisions have been highlighted in red in the revised manuscript and our detailed point-by-point responses are given in the following pages. With these revisions, we hope the anonymous Reviewers will find that the review comments are sufficiently addressed and the current paper is acceptable for publication in Electronics.

With our sincere thanks once again.

Best regards,

All authors

Reviewer 2 Report

Comments and Suggestions for Authors

the paper have been inproved 

Comments on the Quality of English Language

minor editing 

Author Response

Dear Reviewers,

Many thanks for appreciating our revised version. We would like to thank the Reviewer for his/her constructive remarks and careful reading of our paper, which were essential in improving the overall presentation and technical quality of the paper.

With our sincere thanks once again.

Best regards,

All authors
